# A Snapshot of the Frontiers of Client Selection in Federated Learning

**Gergely Dániel Németh**                                   *gergely@ellisalicante.org*
*ELLIS Alicante*

**Miguel Ángel Lozano**                                          *malozano@ua.es*
*University of Alicante*

**Novi Quadrianto**                                      *n.quadrianto@sussex.ac.uk*
*University of Sussex*

**Nuria Oliver**                                            *nuria@ellisalicante.org*
*ELLIS Alicante*

**Reviewed on OpenReview:** *https://openreview.net/forum?id=vwOKBldzFu*

## Abstract

Federated learning (FL) has been proposed as a privacy-preserving approach in distributed machine learning. A federated learning architecture consists of a central server and a number of clients that have access to private, potentially sensitive data. Clients are able to keep their data in their local machines and only share their locally trained model's parameters with a central server that manages the collaborative learning process. FL has delivered promising results in real-life scenarios, such as healthcare, energy, and finance. However, when the number of participating clients is large, the overhead of managing the clients slows down the learning. Thus, client selection has been introduced as an approach to limit the number of communicating parties at every step of the process. Since the early naïve random selection of clients, several client selection methods have been proposed in the literature. Unfortunately, given that this is an emergent field, there is a lack of a taxonomy of client selection methods, making it hard to compare approaches. In this paper, we propose a taxonomy of client selection in Federated Learning that enables us to shed light on current progress in the field and identify potential areas of future research in this promising area of machine learning.

## 1 Introduction

Federated Learning (FL) is a recently proposed machine learning approach that aims to address privacy and security concerns in centralized machine learning. It consists of a collaborative, distributed learning architecture, where a *server* communicates with many *clients* such that the clients keep their potentially sensitive, private data locally and only share the processed model weights and metadata information with the server. The core of the central server's task is to aggregate the model parameters that have been received from the clients to learn an improved global model that is then shared back with the clients. The central server might choose different termination conditions for the training process and perform *model aggregation* using different strategies and optimizers.

Since its proposal in 2017 by McMahan et al. (2017), the FL field has grown very rapidly. A recent comprehensive survey describes the advances and open problems in FL and collects more than 500 related works (Kairouz et al., 2021). This paper follows the same notation and definitions as those presented in Kairouz et al. (2021).

To date, two types of federated learning have been proposed in the literature: First, *vertical federated learning*, where clients have access to *different* data about the *same* individuals, who are related by means of a unique identifier. The motivation for the collaboration in this setting is to build more accurate models by including complementary information about the individuals while preserving their privacy and avoiding sending data to the central server. Second, *horizontal federated learning*, where the features are the same in each client. In this case, the motivation for the collaboration is to have access to more data points thanks to the federation while keeping them privately on the clients' side. The central server learns a better performing model than the local models of any of the individual clients and sends it back to the clients so they benefit from the federation. In this paper, we focus on *horizontal* federated learning techniques.

Regarding the nature of the clients, there are two distinguishable types of FL architectures: *cross-silo* and *cross-device* (Kairouz et al., 2021). In cross-silo federated learning, clients are expected to be reliable, available, stateful, and addressable. Conversely, in cross-device federated learning the clients are separate and diverse individual actors that may not participate in the federation for a variety of reasons, such as a loss of connectivity or excessive energy consumption.

When a federated learning scenario involves few clients, it is feasible to incorporate their parameters in every training round. However, as the number of clients increases, so does the communication overhead, such that considering the data from all the clients becomes a challenge. At the same time, when the number of clients is large, some of the clients might have access to redundant, noisy or less valuable data than other clients. Therefore, *client selection* methods are introduced to reduce the number of working clients in each training round. Recent works have shown that client selection methods are able to keep the performance of the overall model while improving the convergence rate of the FL training (Nishio & Yonetani, 2019), reducing the number of required training rounds (Goetz et al., 2019), or enhancing fairness in case of imbalanced data (Li et al., 2020).

While implementing sophisticated client selection methods may help to achieve these goals, they require the server to have information about the clients, such as training time (Nishio & Yonetani, 2019) or communication stability (Zhou et al., 2021). Thus, the use of client selection methods might have privacy implications, representing a trade-off between potentially losing privacy and achieving good performance while keeping the overhead low, improving the overall utility (Dennis et al., 2021; Wang et al., 2020) or the fairness (Mohri et al., 2019; Li et al., 2020) of the system.

In recent years, different client selection methods have been proposed in the literature (Chen et al., 2018; Mohri et al., 2019; Zeng et al., 2020) with a wide range of objectives, requirements, and experimental settings. Such wealth of proposals in such a short timeframe makes it hard for researchers to properly compare results and evaluate novel algorithms. Furthermore, it limits the ability of practitioners to apply the most suitable of the existing FL techniques to a specific real-world problem, because it is not clear what the state-of-the-art is in the scenario required by their application. The purpose of this paper to fill this gap by providing a comprehensive overview of the most significant approaches proposed to date for client selection in FL.

Our contributions are two-fold: First, we present a taxonomy of FL client selection methods that enables us to categorize existing client selection techniques and propose it as a framework to report future work in the field. Second, we identify missing gaps in existing research and outline potential lines of future work in this emergent area.

The structure of the rest of the paper is as follows: in section 2 we summarize the problem formulation and the mathematical notation used in the manuscript. Section 3 describes the proposed taxonomy and identifies future research directions. Commonly used benchmark datasets are presented in section 4, followed by our conclusions.

## 2 Problem formulation and notation

Federated Learning is a cooperation of N clients, $\mathbb{K} = c_1, ... c_N$, where each client $c_i$ has access to a dataset $\mathbb{D}_i$ which is considered to be private to the client. The clients work together under a central server $S$ to train a global model $f(\boldsymbol{\theta})$ with model parameters $\boldsymbol{\theta}$. The central server aims to minimize the global cost function

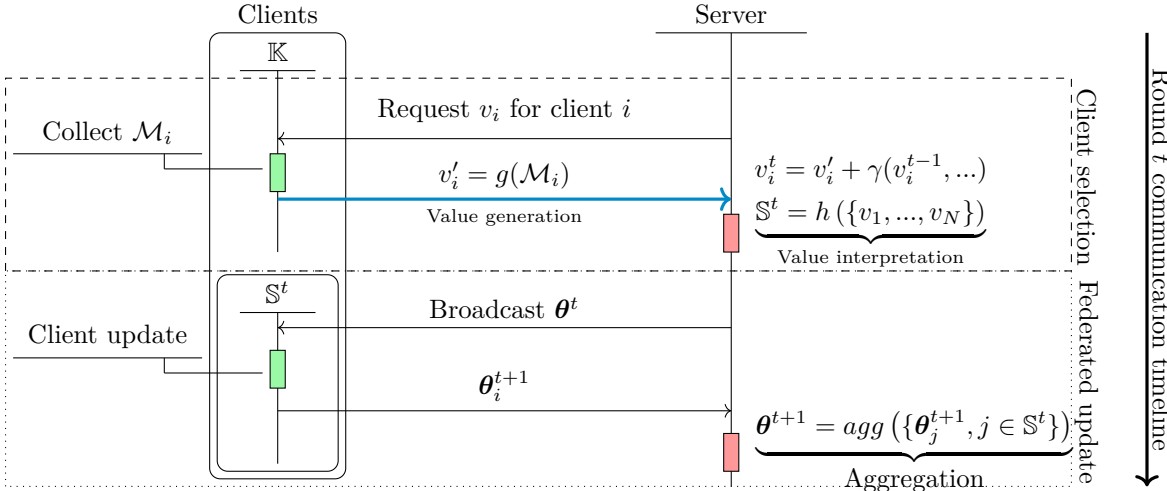

Figure 1: Diagram of a general federated learning training round with client selection. The server communicates with the client set $\mathbb{K}$ to select the participating client set $\mathbb{S}$. Only this subset of clients trains on their local data in a given round $t$ and reports their parameters back to the server. Not all steps are required for every algorithm

$L(\boldsymbol{\theta})$ given by Equation 1. Here $N$ is the total number of clients, $\mathbb{D} = \bigcup_{i=1}^{N} \mathbb{D}_i$ and $|\mathbb{D}|$ is the total number of data samples in all the clients, $(x, y) \in \mathbb{D}_i$ is an input-output pair of training samples in the dataset of the $i$th client and $l$ is the loss function.

$$\min_{\boldsymbol{\theta}} L(\boldsymbol{\theta}) = \frac{1}{|\mathbb{D}|} \sum_{i=1}^{N} \sum_{(x,y) \in \mathbb{D}_i} l(y, f(x, \boldsymbol{\theta})) \tag{1}$$

However, the central server has no access to the clients' private data. Thus, it broadcasts the global model parameters $\boldsymbol{\theta}$ to all clients which are used by each client $c_i \in \mathbb{K}$ as the starting point to compute the local parameters $\boldsymbol{\theta}_i$ that minimize their local cost function $L_i(\boldsymbol{\theta}_i, \mathbb{D}_i)$ described in Equation 2.

$$\min_{\boldsymbol{\theta}_i} L_i(\boldsymbol{\theta}_i, \mathbb{D}_i) = \frac{1}{|\mathbb{D}_i|} \sum_{(x,y) \in \mathbb{D}_i} l(y, f(x, \boldsymbol{\theta}_i)) \tag{2}$$

Once the server has collected the local parameters $(\boldsymbol{\theta}_i)$, it aggregates them into the next iteration of the global parameter $\boldsymbol{\theta}$ in order to minimize the global cost function $L(\boldsymbol{\theta})$. Each iteration of the global parameter update is called a training round and the parameters in the round $t$ are denoted with $\boldsymbol{\theta}^t$. A naive aggregation function is to weight each client's parameter proportionate to their data sample size, given by Equation 3. The federated training proceeds until reaching convergence of the global cost function $L$.

$$\boldsymbol{\theta}^{t+1} = \sum_{i=1}^{N} \frac{|\mathbb{D}_i|}{|\mathbb{D}|} \boldsymbol{\theta}_i^t \tag{3}$$

The communication with each client after each local model update may generate a large overhead, particularly when the number of clients is large. To address this problem, McMahan et al. (2017) introduced the FedAvg algorithm where the clients communicate with the server only after $e$ local epochs. From the server's point of view, the global model parameters are updated with the clients' data in training rounds $t = 1, ..., T$. From the clients' perspective, every training round $t$ consists $\tau = 1, ..., e$ local epochs. Thus, the clients train a total number of $eT$ epochs.

Additionally, FedAvg uses a random selection $\mathbb{S}^t$ of $M$ participating clients in every training round $t$, $\mathbb{S}^t \subseteq \mathbb{K}$ with $|\mathbb{S}^t| = M$, such that a client $i$ is selected with probability $p(c_i \in \mathbb{S}^t) = \frac{M}{N}$. The client update and aggregation steps are given by $\boldsymbol{\theta}_j^{t+1} = ClientUpdate(\boldsymbol{\theta}^t, \mathbb{D}_j)$ and $\boldsymbol{\theta}^{t+1} = agg(\{\boldsymbol{\theta}_j^{t+1}, j \in \mathbb{S}^t\})$, respectively.

After the seminal work of FedAvg, subsequent research works have proposed a diversity of client selection methods, most of which follow the flow illustrated in Figure 1. First, the server requests a value $v_i$ from each client, which the client computes based on its local information $\mathcal{M}_i$. Once the server has received the $v_i$ from all the clients, it selects a subset $\mathbb{S}^t$ of the clients according to function $h(v_1, ..., v_N)$. Stateful selection methods also use previous information about the clients, calculating $v_i$ from the client feedback $v_i'$ and a function ($\gamma$) of previous values of the client. After this, the server broadcasts its global parameters $\theta^t$ only to the selected clients $\mathbb{S}^t$ which update their local models and send back their updated local parameters $\theta_i^{t+1}$. With this information, the server computes the updated global parameter $\theta^{t+1}$.

Based on this general description of client selection in federated learning, we present next the proposed taxonomy to characterize client selection methods.

## 3 Proposed taxonomy of client selection methods

Our proposed taxonomy, depicted in Figure 2, aims to ease the study and comparison of client selection methods. It contains six dimensions that characterize each client selection method. The top part of the Figure displays the three dimensions (marked in red) that concern the server in the Federated Learning framework whereas the bottom three dimensions concern the clients (marked in green) or both the clients and the server (marked in blue). In the following, we describe in detail each of these dimensions and present the most relevant works in the client selection literature in FL according to the proposed taxonomy.

### 3.1 Client Selection Policies

The main purpose of client selection in Federated Learning is to automatically determine the number of working clients to improve the training process. A broad range of policies have been proposed in the literature, including a variety of global constraints (e.g. improving the efficiency in the learning, limiting the amount of time or computation needed or ensuring fairness), pre-defined client inclusion criteria and client incentives.

We provide below a summary of the most prominent works according to their policies for client selection. The third column in Table 1 provides an overview of the client selection policies of the most representative client selection methods in FL.

#### 3.1.1 Global Constraints

The server might define different types of global constraints to drive the client selection process.

**Training efficiency**

The most common goal in client selection is to *speed up the convergence* of the training process. However, formalizing this simple goal may result in different technical solutions. Some authors (Chen et al., 2018; Dennis et al., 2021) focus on reducing the number of clients in a training round while maintaining the same convergence rate. Chen et al. (2018) show that communicating only with the right clients can reduce the communication to the tenth of what it would be in a cyclic iteration of the clients. Later work by Chen et al. (2020) reports that optimal client sampling may yield similar learning curves to those in a full participation scheme. Other works aim to reduce the total number of training rounds to reach the same accuracy. Cho et al. (2022b) propose pow-d which takes half as many iterations to reach 60% accuracy on the FashionMNIST dataset than a random client selection.

Alternative definitions of efficiency include energy consumption. For example, Cho et al. (2022a) propose the FLAME$_C$ FL framework that computes energy profiles for each client and simulates different energy profiles in the federation. They report that incorporating energy consumption as a global constraint in the

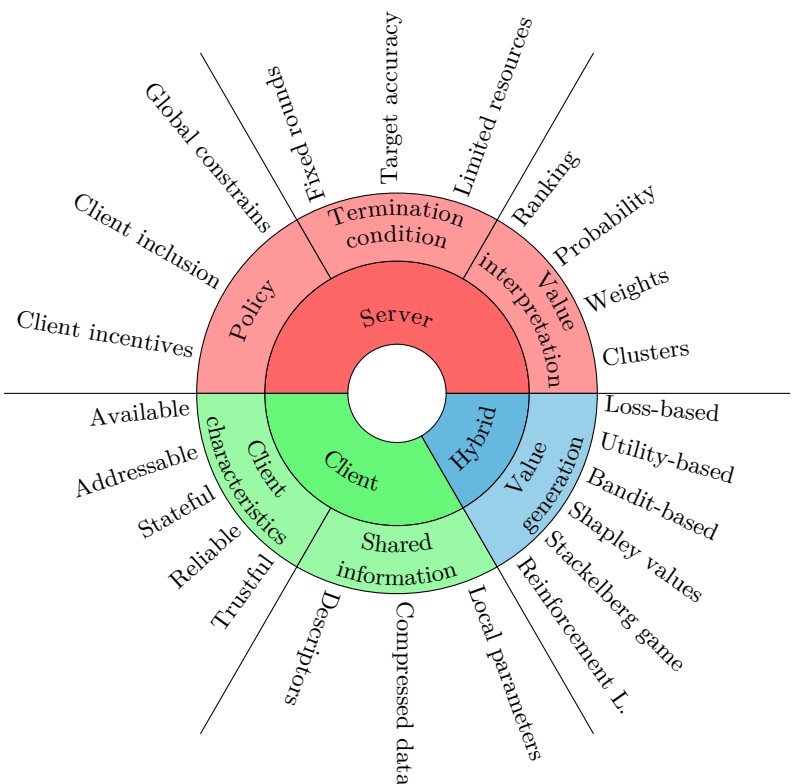

Figure 2: Proposed taxonomy of client selection methods in Federated Learning. The taxonomy has six dimensions: Policy, Termination Condition, Value Interpretation, Client Characteristics, Shared Information, and Value Generation. The top three dimensions (colored in red) capture properties on the server side, the two dimensions colored in green are on the client side, and the sixth dimension, colored in blue, corresponds to a hybrid client-server property.

client selection method can save up to 2.86× more clients from reaching their energy quota when compared to the same method without any energy consideration.

Other authors propose limiting the time of the local training as the main guiding principle to achieve client selection. This concept was introduced by Nishio & Yonetani (2019) in the FedCS method, which filters out slow clients and hence speeds up the overall training time. FedCS requires clients to estimate $\mathcal{T}_i$, the time needed to perform their round of learning and the server selects only clients with $\mathcal{T}_i$ lower than a certain threshold. The authors report that selecting the *right* clients according to this criterion could reduce by half the total training time to reach the desired accuracy when compared to federated average (FedAvg) with limited local training round time.

**Resilience**

Client selection may be used to remove malicious clients from the training process, thus increasing the resilience of the system against adversarial attacks. Rodríguez-Barroso et al. (2022) use a server-side validation set $\mathbb{D}_V$ to measure the accuracy of each local model. Then, the models with low accuracy are given lower weights. Their results show that this method is able to keep the same level of performance even if 10 out of 50 clients send malicious model updates. An alternative approach to filter malicious clients is using *Client Incentives*. In these methods, the rewards offered to the clients are inversely proportional to the probability of being a malicious or harmful client. We direct the reader to section 3.1.3 for a description of such methods.

While client selection can be a tool to fight against attacks, it opens a new vulnerability. Malicious clients may attempt to be included in the training rounds by changing their behavior or their data to be more appealing for selection. For example, the method proposed by Blanchard et al. (2017) which selects clients with the smallest weight update is vulnerable to adaptive attacks. It could be abused by injecting a backdoor with a very small learning rate, and therefore small weight update. The server would then be selecting this client over other benign clients.

**Fairness**

An additional policy for client selection is to ensure **fairness** in the criteria applied to select clients. In FL, and especially in cross-device FL, algorithmic fairness definitions, particularly group fairness definitions adopted in the machine learning literature are hard to implement because they require knowledge of sensitive attributes that are only available on the clients' side. For example, providing group fairness guarantees according to a protected attribute (e.g. race or gender), would require the server to collect data of the performance of the model on different groups according to the protected attribute. However, the privacy motivation for FL would prevent such attributes from being shared with the server. One possible way to report sensitive data with privacy protection would be to use *differential privacy* (Geyer et al., 2017; Padala et al., 2021).

In the context of FL, different definitions of *fairness* have emerged. Work by Mohri et al. (2019) aims to increase good-intent fairness to minimize the maximum loss in the federated training between clients. In the experimental evaluation, the proposed $AFL_M$ method outperforms the baseline uniform distribution in various datasets while increasing the worst-case accuracy, i.e. the performance on the client with the lowest accuracy. This principle is related to min-max notions of fairness, such as the principle of distributed justice proposed by Rawls (2004).

Li et al. (2020) define fairness as the property of achieving *uniform accuracy* across all clients. This definition corresponds to the parity-based notion of fairness, as opposed to the min-max principle described above. They propose a novel FL approach called q-FFL to reduce potential accuracy differences between clients by giving a larger weight to those clients with a large error during the training process. According to their experiments, the proposed method yields an increase of the worst accuracy by 3% in a class-split Fashion-MNIST federated experiment while keeping the average accuracy the same as a state-of-the-art method ($AFL_M$).

Shi et al. (2021) propose a taxonomy to categorize the fairness definitions that have been proposed in the FL literature. All the proposed fairness definitions belong to the Server Policies dimension of our taxonomy (see Figure 1): *accuracy parity* and *good-intent fairness* would be categorized as Global Constraints; *selection fairness* could also be considered a Global Constraint or part of a Client Inclusion policy; finally, *contribution, regret distribution* and *expectation fairness* would belong to the Client Incentives policy.

### 3.1.2 Client inclusion policies

In some cases, the server might define specific client inclusion policies beyond the global constraints. For example, if a client has limited, but very relevant or valuable data to the problem at-hand (e.g. data from an underrepresented demographic group), the server may define client inclusion policies to ensure that such clients are selected even if they would not satisfy the global constraints.

Examples of client inclusion policies include ensuring a **loss tolerance** for communication packages. Zhou et al. (2021) show that relaxing the global constraints for clients with poor communication channels not only improves client inclusion but also the overall performance of the FL system. Their proposed method LT-FL allows clients to discard lost packages instead of asking the server to resend them. Thus, the communication time decreases and otherwise poorly-represented clients are able to participate in the training.

An alternative client inclusion policy may be seen in the context of **model-agnostic FL**. In this case, low-resource clients have the option to learn different, simpler models. Thus, the local training may be faster and the communication requires less data transfer. One example of such an inclusion policy is Federated Dropout (FD) proposed by Caldas et al. (2018b), where simpler deep neural networks (with fewer neurons in

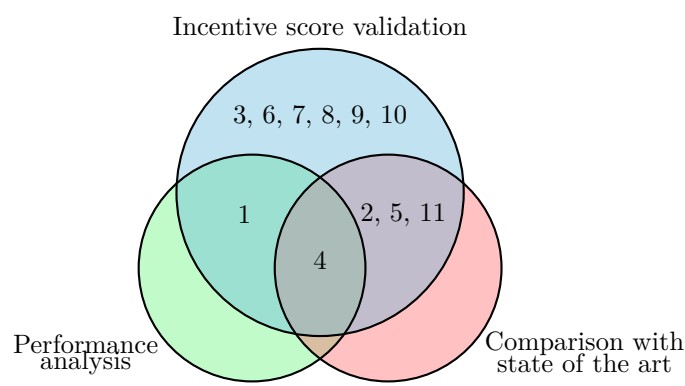

Incentive score validation

3, 6, 7, 8, 9, 10

1

2, 5, 11

4

Performance analysis

Comparison with state of the art

Highlighted Incentive Mechanisms for FL Client Selection

1. Zeng et al. (2020)
2. Kang et al. (2019a)
3. Sarikaya & Ercetin (2019)
4. Zhao et al. (2020)
5. Song et al. (2019)
6. Wang et al. (2019a)
7. Liu et al. (2020)
8. Pandey et al. (2019)
9. Hu & Gong (2020)
10. Lim et al. (2020)
11. Kang et al. (2019b)

Figure 3: Experimental evaluations of proposed incentive mechanisms in the literature lack a quantitative comparison with state-of-the-art methods on benchmark datasets with a comprehensive performance and efficiency analysis.

the layers) are learned in clients and the server combines them together into a larger network following the idea of dropout. Their method shows a 14× reduction in server-to-client communication, a 1.7× reduction in local computation and a 28× reduction in client-to-server communication compared to uncompressed models.

Following this work, Diao et al. (2020) and Liu et al. (2022) propose methods to match weaker clients with smaller models and implement a model-agnostic FL. In HeteroFL, proposed by Diao et al. (2020), the smaller model's parameter matrix is a cropped version of the complete matrix. During the aggregation, the server aggregates the models with different sized hidden layers from smaller to larger. They present experimental results with 5 different sized models, the largest having 250 times as many parameters as the smallest. They report similar accuracies when setting half of the clients' models to the smallest size and half to the largest size than when all the clients have the largest model while halving the average number of parameters. Liu et al. (2022) propose InclusiveFL where larger models have more hidden layers. They show that for more complex models (e.g. transformer models with the self-attention mechanism) InclusiveFL works better than HeteroFL on several ML benchmark datasets.

### 3.1.3 Client incentives

A common assumption in Federated Learning is that all the clients are willing to participate in the training at any time. The reward for their contribution to the federation is access to an updated, global –and ideally better– model. However, clients might not see the benefit of the federation for a variety of reasons. For example, stateless clients in cross-device FL may never receive an updated model from the server after participating because the users might disconnect their devices before being selected a second time. Thus, an active research area in the literature of client selection in FL focuses on defining *incentive mechanisms* for the clients so they participate in the federation. Note that such incentive mechanisms have been extensively studied in other multi-actor areas of science, such as economy (Hahn & Stavins, 1992; Lindbeck et al., 1999), environmental protection (Nichols, 1984; Kremen et al., 2000), mobile crowdsensing (Yang et al., 2017), and shared distributed energy sources (Wang et al., 2019b) or human resources (Farzan et al., 2008).

In the context of FL, the server typically gives a reward to the participating clients according to their contribution. According to the work by Zhan et al. (2021), the incentive mechanism methods proposed to date may be categorized into three classes, described below. Zeng et al. (2021) additionally categorize the methods based on the used value generation techniques, described in Section section 3.5.

The first class of incentive mechanisms considers **client resource allocation** and gives rewards based on the computation power, energy usage, or allocated time. For example, Sarikaya & Ercetin (2019) offer a Stackelberg game-based solution based on the clients' CPU power utilization. They study the server's

strategy based on its budget and the trade-off between the number of learning rounds and the required time of a round with respect to the number of clients.

Second, the incentive mechanisms may be used to attract clients with good data quality and quantity. These incentive mechanisms are classified as **data contribution** incentives. For example, Song et al. (2019) use Shapley values (Shapley, 1971) to evaluate the data contribution of clients and reward them accordingly. They define the contribution index (CI) such that (1) data distributions that do not change the model have no contribution; (2) two datasets with the same influence on the model have the same CI, and (3) if there are two disjoint test sets, the contribution index in the context of the union test set should be equal to the sum of the CI of the two original test sets. The proposed CI-MR method performs similarly to state-of-the-art methods while calculating the scores up to 5 times faster. Zeng et al. (2020) show that measuring the data contribution by means of a clients' utility score and applying an auction, the FMore method selects 80% of the participating clients, resulting in a $40 - 60\%$ speed-up in terms of the number of training rounds necessary to reach 90% accuracy on the MNIST dataset.

Third, the incentives may be correlated with the **reputation** of the clients. Such reputation may be computed from the usefulness of each of the client's models. In addition, several works have proposed using a blockchain to keep track of the clients' contributions and hence of their reputation (Kang et al., 2019a; Zhao et al., 2020). Kang et al. (2019a) show that removing clients with a bad reputation may increase the model's final accuracy. Zhao et al. (2020) report that a reputation score may punish potentially malicious clients while maintaining the global convergence of the model.

**Future Work**

There are several areas of future work regarding the Policies of the server to perform client selection.

We refer to the lack of discussion on privacy-efficiency trade-off more in Section 3.4. Here we want to highlight work on privacy as a Global Constraint. Geyer et al. (2017) define a privacy budget for each client and propose an experiment where clients terminate their participation if their privacy budget is reached. Thus, clients terminate in different rounds. This line of work can be expanded to more complex budgeting that reflect the information shared in messages.

While many client selection approaches have been proposed in the literature, they lack a vulnerability analysis of potential attacks directed against the approach itself. Future work in this direction would be needed before a specific client selection method is deployed in production.

Another potential research direction would focus on the clients' perspective to define the client selection strategy. While client incentive policies aim to motivate the clients' participation with fair payoffs, there is an opportunity to further study and propose FL approaches centered around the clients' interests. In this sense, we identify several relevant research questions, such as studying the trade-off between data availability and performance of the local vs federated learning models; developing models to support the clients' decisions as to when to join the federated learning scenario; and approaches that would enable clients to participate in the federation in a dynamic manner, depending on the current state of the learning model. Recent advances in model-agnostic FL in combination with incentive mechanisms suggest an interesting direction of future work. One could design a federated learning scenario, where the server would propose contracts to the clients with different model sizes and rewards. Then, the clients would be able to select the contract that matches their available resources and interests the best.

Finally, we believe that there is an opportunity to improve the empirical evaluation of existing and newly proposed incentive mechanisms for FL. There are 3 key components –depicted in Figure 3– that we consider necessary to include in such an evaluation: first, a clear explanation as to why the proposed scoring satisfies the requirements of being an incentive mechanism; second, a performance and efficiency analysis on benchmark datasets; and third, a systematic comparison with other state-of-the-art client selection methods. As can be seen in the Figure, most of the proposed methods in the literature fail to include one or more of these three elements. While several survey papers of incentive mechanisms in Federated Learning have been published (see e.g. Zhan et al. (2021); Zeng et al. (2021)), to the best of our knowledge none of them

presents a thorough quantitative analysis of the topic. In section 4 we further discuss the scarcity of available benchmark datasets and experimental setups for Federated Learning.

## 3.2 Termination condition

The next dimension in the proposed client selection taxonomy is the Termination condition, which is defined by the server. The termination condition determines when to end the Federated Learning training process, and hence it is an important choice of the experimental setup. The definition of the termination condition typically depends on the goal and priorities of the FL system. The seventh column in Table 1 provides an overview of the termination conditions of a sample of the most representative client selection methods proposed to date in the FL literature.

We identify three main categories regarding the termination condition.

In the first category of methods, the learning process ends after a **fixed number of rounds** ($T$) of federated training. In simulated scenarios, this termination condition helps to identify whether different methods reach the same accuracy. However, in some works the methods do not reach their best performance in a predefined number of rounds (Chen et al., 2020; Cho et al., 2022a). In this case, it is important to motivate why the experiments ended at that point. For real-world scenarios, this termination condition is easy to implement.

In the second category of approaches, the learning process ends when a **minimum required accuracy** is achieved. This termination condition is particularly helpful to compare different federated learning methods regarding how quickly they achieve the desired accuracy. The minimum required accuracy may be determined by the performance of a baseline method or by the task at hand to solve with the federated learning system (Nishio & Yonetani, 2019; Zeng et al., 2020; Lai et al., 2021).

Finally, alternative termination conditions may be defined from the perspective of **limiting the participation of a client**: for example, Cho et al. (2022a) limit the energy drain a client's device is allowed to endure. Furthermore, Kang et al. (2019a) limit the reward clients can receive in an incentive mechanism, thus the experiment finished when all the rewards are spent. If the server has a validation set, it may apply an early stopping ($E_S$) approach. Otherwise, the client may suggest that they are unable to meaningfully contribute to produce a better model. In (Chen et al., 2018), the proposed LAG-WK FL system allows the clients to evaluate whether they should send an update to the server or not, and the training stops when no client sends a new round of parameters.

### Future work

In terms of future work regarding the termination condition, we highlight two potential directions. First, privacy could be considered as a key factor to define the termination of the learning process (Geyer et al., 2017). The PyShift package, designed by Ziller et al. (2021) was implemented with this consideration in mind, giving a privacy budget to the clients which is reduced every time the server accesses information from the specific client. Beyond PyShift, we identify two potential lines of future work: an analysis of the potential privacy loss of the clients that participate in several training rounds and the inclusion of a participation budget as a global constraint to protect the privacy of the clients' data.

Second, current works give the right of termination to the server. However, in a real-life scenario, the clients could also decide to stop their participation in the training process. Incentive mechanisms aim to achieve a fair client participation, but they still give the power of termination to the server and generally do not consider the clients' interests or perspectives. Client-centric termination policies could therefore be a potential line of future work.

## 3.3 Client characteristics

FL methods are implemented with certain assumptions regarding the characteristics that the clients should have. Thus, the Client Characteristics are the next dimension in the proposed taxonomy. The second column in Table 1 provides an overview of the client characteristics required by a representative sample of the client selection methods proposed to date in the FL literature.

Table 1: Selected works from the literature and their characteristics according to the proposed taxonomy. Appendix A shows the detailed descriptions of the methods. Client characteristics: availability ($V$), addressability ($D$), statefulness ($S$), reliablity ($R$), and trustworthiness ($T$). $\mathcal{E}$ indicates an exploration term with additional randomly selected clients and $E_S$ indicates an early stopping mechanism.

| Algorithm | Client characteristics | Policy | Message | Value generation | Value interpretation | Termination condition |
|---|---|---|---|---|---|---|
| EqRep$_{\text{baseline}}$ | $V\overline{DS}RT$ | Selection fairness | $\boldsymbol{\theta}$ | - | Probability | - |
| EqW$_{\text{baseline}}$ | $VD\overline{S}RT$ | Selection fairness | $\boldsymbol{\theta}, |\mathbb{D}|$ | - | Probability | - |
| AFL$_{\text{G}}$ | $\overline{V}DSRT$ | Efficiency | $\boldsymbol{\theta}, L$ | Loss-based | Probability+$\mathcal{E}$ | Fixed rounds |
| pow-d | $\overline{V}D\overline{S}RT$ | Efficiency | $\boldsymbol{\theta}, |\mathbb{D}|, L$ | Loss-based | Ranking | Reach accuracy |
| S-FedAvg | $VDSRT$ | Efficiency | $\boldsymbol{\theta}$ | Shapley values Validation set | Probability | Fixed rounds |
| $k$-FED | $VDSRT$ | Efficiency | $\boldsymbol{\theta}, \mathbb{G}$ | Clustering | Clusters | Fixed rounds |
| LAG | $VDSRT$ | Efficiency | $\boldsymbol{\theta}, \mathcal{L}$ | Loss-based | Threshold | Fixed rounds,$E_S$ |
| FL-CIR | $VDSRT$ | Efficiency | $\boldsymbol{\theta}, L$ | Bandit-based | Ranking | Fixed rounds |
| FAVOR | $VDSRT$ | Efficiency | $\boldsymbol{\theta}$ | Reinforcement L. Validation set | Ranking | Reach accuracy |
| FedCS | $\overline{V}D\overline{S}RT$ | Limit round time | $\boldsymbol{\theta}, \mathcal{T}$ | Time-based | Ranking | Fixed rounds, Reach accuracy |
| AFL$_{\text{M}}$ | $VDSRT$ | Good-intent fairness | $\boldsymbol{\theta}, L$ | Loss-based | Probability | Fixed rounds |
| q-FFL | $\overline{VDSR}T$ | Uniform accuracy | $\boldsymbol{\theta}, \mathcal{L}$ | Loss-based | Weights | Fixed rounds |
| Oort | $\overline{V}DSR\overline{T}$ | Limit round time | $\boldsymbol{\theta}, L, \mathcal{T}$ | Utility game | Ranking+$\mathcal{E}$ | Reach accuracy |
| FLAME$_C$ | $\overline{V}DSRT$ | Limit client participation | $\boldsymbol{\theta}, L, \mathcal{T}, E$ | Utility game | Ranking | Limited energy |
| DDaBA | $\overline{VDS}R\overline{T}$ | Defense against attacks | $\boldsymbol{\theta}$ | Validation set | Weights | Fixed rounds |
| LT-FL | $\overline{V}D\overline{SR}T$ | Client inclusion | $\boldsymbol{\theta}, \mathcal{T}$ | Time-based | Clusters | Fixed rounds |
| FD | $\overline{V}D\overline{S}RT$ | Client inclusion | $\boldsymbol{\theta}$ | Model compression | Weights | Fixed rounds |
| CI-MR | $VDSRT$ | Client incentives | $\boldsymbol{\theta}, |\mathbb{D}|$ | Shapley values | Weights | Fixed rounds |
| FMore | $\overline{V}D\overline{S}RT$ | Client incentives | $\boldsymbol{\theta}, |\mathbb{D}|, \mathcal{T}...$ | Auction Utility score | Ranking | Fixed rounds, Reach accuracy |
| CBIM | $\overline{V}DS\overline{RT}$ | Client incentives | $\boldsymbol{\theta}, \mathcal{T}, E...$ | Contract theory Utility score | Threshold | Limited reward |

According to Table 1 from Kairouz et al. (2021), FL clients may have four important characteristics: availability, addressability, statefulness and reliability. We propose to add a fifth characteristic, *trustworthiness*, that should be considered when designing a client selection method in a FL scenario. These five characteristics are defined as follows:

**Availability** ($V$ or $\overline{V}$): Let $\mathbb{A}$ be the active client set, i.e. the clients that are ready to participate in the next training round. In each training round, $t$, the server may select the participating clients $\mathbb{S}^t$ from the active client set $\mathbb{A}^t$. Generally, $\mathbb{S}^t \subseteq \mathbb{A}^t \subseteq \mathbb{K}$. Sometimes, for example in cross-silo scenarios, $\mathbb{A}^t = \mathbb{K}, \forall t = 1...T$. The assumption of client availability in a FL method will be denoted by $V$, otherwise $\overline{V}$.

**Addressability** ($D$ or $\overline{D}$): Clients have unique identifiers, such that the server is able to individually address them and communicate with specific clients ($D$). Conversely, the server communicates in a broadcasting manner, sending the same message to all the clients ($\overline{D}$).

**Statefulness** ($S$ or $\overline{S}$): In a stateful scenario ($S$), the clients are able to participate in multiple training rounds and they can refer back to the weights or parameters of previous rounds. In a stateless scenario ($\overline{S}$), each client participates only once in the learning process, so the client selection (and aggregation) policy is not able to depend on their previous performance.

**Reliability** ($R$ or $\overline{R}$): Even if a client meets all the selection criteria and is selected by the server to participate in a round of training, it may fail before sending its information to the server. Federated learning methods that assume that all participating clients will successfully report back at the end of their local training make a reliability assumption, $R$. Otherwise, the methods are characterized as $\overline{R}$.

**Trustworthiness** ($T$ or $\overline{T}$): In an ideal scenario, all clients are sharing truthful information with the server and hence they are trustworthy, $T$. However, clients may be malicious ($\overline{T}$) and share false information. Previous research in FL has focused on tackling this problem, such as Kang et al. (2019a) who propose a client selection incentive that gives a bad reputation score to potentially harmful clients and Rodríguez-Barroso et al. (2022) who gives less weight to clients with lower accuracy on a server-side validation set.

In general terms, these five characteristics of the clients are known to the server in cross-silo FL scenarios. However, in cross-device scenarios, several of these features may not be known to the server and the methods should be able to function without depending on any of them. We would like emphasize the importance of new FL client selection methods to identify their dependencies on these client characteristics such that different approaches may be compared and the reproducibility of the results is facilitated.

**Future work**

Client characteristics are not always considered when designing client selection policies which leads of inefficient or inconsistent FL systems. For example, cross-device FL is in many cases stateless as clients may join to the system only once during the training. However, there are reputation-based client incentives (based on the client's performance in previous rounds) proposed for cross-device FL that are impossible to implement as the server does not have access to the clients' state in a stateless context (Kang et al., 2019a). The same problem was identified by Wang et al. (2021) for FL in general. We would like to emphasize the importance of clearly describing the required client characteristics when proposing novel approaches to FL to ease the comparison with other methods and ensure the reproducibility of the results.

Trustworthiness in FL is often addressed at the level of optimization algorithms or model aggregation. Blanchard et al. (2017) propose an algorithm to weigh the client models during aggregation in a way to reduce the effect of outliers. However, few research works address trustworthiness during client selection by e.g. removing harmful clients Zhao et al. (2020). We believe the addition of this feature to the client characteristics in our taxonomy may boost future research in this direction.

In addition to Client Characteristics, the proposed taxonomy includes a dimension related to the kind of information that the client shares with the server.

### 3.4 Shared information by the client

Federated learning was proposed with the motivation of preserving privacy by enabling a central server to learn from distributed client data without ever having access to the actual data. As the learning heavily depends on the information shared by the clients with the server, it is important to understand and analyze the types of messages that the clients send to the server.

Client selection requires analyzing information about the clients. Thus, there is a trade-off between a potential privacy loss from the client's perspective and the goals of implementing client selection. This section of the proposed taxonomy highlights the importance of studying and characterizing such a trade-off.

The most commonly shared client information are either the updated client weights ($\boldsymbol{\theta}_i$) or the change in the weights when compared to the previous learning round (Nagalapatti & Narayanam, 2021; Wang et al., 2020). Additional methods proposed in the literature use the loss ($L$) of the clients' model (Goetz et al., 2019; Mohri et al., 2019; Yang et al., 2021).

Moreover, clients might share additional information with the server. Examples from the literature may be categorized into two groups. Firstly, **scalar descriptors**, such as the Lipschitz-smoothness of the client loss function ($\mathcal{L}$) (Chen et al., 2018; Li et al., 2020), the total size of the client data ($|\mathbb{D}|$) (Song et al., 2019; Cho et al., 2022b; Zeng et al., 2020) or the expected training time on the client ($\mathcal{T}$) (Nishio & Yonetani, 2019; Kang et al., 2019a; Zeng et al., 2020; Zhou et al., 2021).

Secondly, a modified, **compressed version of the client data**, such as a vector $\mathbb{G}$ of cluster centers computed from the clients' data (Dennis et al., 2021).

The message sharing between the client and the server is denoted by $v_i = g(\mathcal{M}_i)$, $i : 1, ..., N$, where $\mathcal{M}_i$ is the shared information by client $i$. To select the best clients, the server needs to assign a value ($v_i$) to each client. The function $g$ generates this value from the client's shared information. This Value Generation process (described in section 3.5 below) may be done on either the server or the client side. Note that a server-side validation process to evaluate the clients' models reduces the server-client communication, but adds extra overhead on the server as it has to run all the client models on the validation set ($D_V$) (Wang et al., 2020; Nagalapatti & Narayanam, 2021).

The fourth column in Table 1 includes the types of Messages send by the clients to the server in each of the selected representative papers from the literature and the fifth column describes the Value Generation functions used by such papers.

To preserve the clients' privacy, several privacy-preserving methods have been used in FL frameworks, such as *differential privacy* Abadi et al. (2016) and *secure aggregation* Bonawitz et al. (2017). Despite a call from Wang et al. (2021) in a general field study to make client selection methods compatible with privacy-preserving methods, current client selection works do not discuss their compatibility. For example, client selection approaches relying on a server-side validation set (see Value generation column in Table 1) cannot apply secure aggregation as they require server-side inference on the local models. Similarly, client selection strategies might not work if local differential privacy is used to disclose the clients' update Duchi et al. (2013).

### Future work

While current works analyze the communication in terms of performance and calculate the trade-off of including additional messages, there is a lack of research analyzing the privacy cost of sharing increased amounts of information between the clients and the server. In addition, future work is needed on the interplay between client selection and privacy-preserving methods, such as differential privacy and secure aggregation.

Using locally generated synthetic data for client selection may be another fruitful research direction. Tackling the challenge of non-identically and independently distributed (IID) distributed data problem has motivated GAN generated synthetic data sharing and progress was made in this direction (Xin et al., 2020; Rajotte et al., 2021), however, these methods do not leverage the client selection, only use all clients or implement random selection.

The next dimension in the proposed taxonomy is Value Generation, i.e. the process by which the server assigns a value to each client that will be used to inform the client selection process.

### 3.5 Value generation

Once the server receives each clients' shared information $\mathcal{M}_i$, it generates a relevance value ($v_i$) for each client. This value is used to select the set of participating clients in the next round of training ($\mathbb{S}^t$). In this section, we summarize the most significant Value Generation approaches proposed in the literature.

While the value generation typically takes place on the server side, in some cases the clients send an already processed value to the server, or the value might be even computed in both sides. We discuss how the server interprets this generated value in section 3.6.

A commonly used technique is to evaluate the clients directly using their local training loss. Intuitively, if a client has a large loss, the training would benefit from more rounds of the client's data. However, these **loss-based** methods need to store the results in previous rounds of the training, requiring either stateful clients (Chen et al., 2018) or clients that are able to estimate their loss in the current round. In this case and to reduce the overhead, some methods use a random selection of clients first ($\mathbb{A} \subset \mathbb{K}$) and then implement a more sophisticated client selection strategy on this subset of clients. In (Cho et al., 2022b), the authors show that optimizing the size of this subset has a positive impact on the model's performance: their pow-d method outperforms EqRep$_{\text{baseline}}$ by 10% and AFL$_{\text{G}}$ by 5% on the FMNIST dataset.

Other research suggests that the loss is only one of different metrics that one could optimize for. Lai et al. (2021) proposes a **utility function** composed of different terms, including the loss-based statistical utility and the system's utility derived from the time needed for the training. They report a significant speed-up in training time: the proposed Oort reaches better accuracy in 10 hours than a random selection method in 30 hours on the OpenImage (Kuznetsova et al., 2020) image classification task. Cho et al. (2022a) incorporates a third utility term: an energy consumption-based score. They show that distributing the workload with respect to the energy usage doubles the number of training rounds without dropping clients due to reaching their energy limit. The general formulation for computing an overall utility score is given by equation 4, where $\mathbb{B}_i \subset \mathbb{D}_i$ is a subset of the $i$th client's data.

$$Util(i,t) = \underbrace{u_1(L_i^t, \mathbb{B}_i)}_{Statistical\ utility} \times \underbrace{u_2(\mathcal{T}_i^t)}_{System\ utility} \times \underbrace{\cdots}_{Other\ utilities} \tag{4}$$

Other scholars have formulated the client selection problem as a **bandit problem**. Yang et al. (2021) propose a multi-armed bandit approach where the clients are the arms and the $\mathbb{S}^t$ client set is the superarm at round $t$ to achieve reduced class imbalance between clients and thus increase the training efficiency. Their method yields a 10% increase in accuracy when compared to random selection in a non-IID scenario of the CIFAR10 dataset.

In cooperative game theory, **Shapley values** are used to determine the contribution of clients or the value of their data (Shapley, 1951). In FL, they are used to give a fair payoff to participating clients based on their contribution (Wang et al., 2019a; Song et al., 2019; Liu et al., 2020) and to identify the most valuable clients for the next training round (Nagalapatti & Narayanam, 2021). Wang et al. (2019a) use Shapley values to determine the features with the largest contribution in a vertical FL scenario. Song et al. (2019) show that Shapley values are effective to compute a contribution index of the clients. Liu et al. (2020) simulate clients with different data quality levels from the MNIST dataset and show that their Shapley value-based method gives higher scores to the clients in higher data quality group. While the above 3 works use Shapley values for incentives, Nagalapatti & Narayanam (2021) show that removing irrelevant clients using Shapley values

can increase the overall accuracy. Their S-FedAvg increases the validation accuracy from 40% to 80% when compared to FedAvg in a MNIST-based experiment.

The **Stackelberg game** is a market modeling structure with two types of actors: leaders and followers. In FL, the server acts as the leader and the clients as the followers. In Sarikaya & Ercetin (2019), the server proposes a price for the clients' resources which the clients use to compute their utility scores that are sent to the server. Finally, the server selects the clients with the highest scores. Their experiments support that there is an optimal number of clients from an efficiency perspective that may be identified with such a client selection approach, despite the availability of more clients. Furthermore, Pandey et al. (2019) propose to reward the clients' local accuracy and show that their method can achieve optimal utility scores in a small, 4-client scenario. Hu & Gong (2020) propose a Stackelberg game to give incentives depending on the clients' data privacy loss. Unfortunately, these works lack both a comparative analysis of their empirical results with other state-of-the-art approaches and a performance analysis on commonly used benchmark FL datasets.

Auction and contract theory have also been proposed to assign a value to clients in FL incentive mechanisms. In this case, the server acts as the auctioneer and the clients bid with their local resources. The FMore system presented in Zeng et al. (2020) uses an auction to select the clients with the best utility-bid pair. The authors report a reduction of $45 - 68\%$ in the training rounds needed to reach a given accuracy in experiments with various datasets. Jiao et al. (2020) demonstrate that the auction-based method can be applied in a FL scenario where clients compete for wireless channels. Deng et al. (2021a) measure the learning quality of clients and apply an auction to select the participants. Their method is robust against attacks and achieves $50 - 80\%$ accuracy in tests where the FedAvg baseline method only achieves 10%.

In **contract theory**, the server proposes rewards for different client types and the clients select the cluster they fit in according to their local resources. Kang et al. (2019b) separate the different types of clients by their local model accuracy. Their work focuses on maximizing the profit that remains at the server after incentive payout. Lim et al. (2020) also define the client groups based on data quality and quantity. Their experiments show that these methods effectively reward the desired clients and motivate them to participate more than the less relevant ones.

Finally, **Reinforcement Learning** (RL) has also been proposed for client selection. Wang et al. (2020) frame the client selection process as a RL task as follows: the server acts as the agent; the state $s^t = \boldsymbol{\theta}^t, \boldsymbol{\theta}_1^t, ..., \boldsymbol{\theta}_N^t$ summarizes the current parameters in each client; the action consists of selecting a client for the next round; and the reward is the sever's side accuracy. In their experiments the proposed FAVOR method reduced the number of communication rounds by $23 - 49\%$ on baseline datasets. Deng et al. (2021b) take the client's data quality into account during value generation and show that their method selects fewer and more suitable clients than a simple baseline, achieving 6% better accuracy while being $2\times$ faster. RL has also been used to implement incentive mechanisms combined with auctions or Stackelberg games. Jiao et al. (2020) implement a RL-based auction that outperforms a greedy auction by $2 - 5\%$. Zhan et al. (2020) propose using reinforcement learning to solve the Stackelberg game problem and approximate the equilibrium.

**Future work**

Regarding utility score-based models, there is a trend to find new, relevant descriptors of the system –such as energy in Cho et al. (2022a), include them in the utility function and show better results than those of previous methods. This trend can be recognized in incentive mechanisms as well, where the reward function is based on the utility of local resources. This line of work suggests a potential direction of future work by systematically collecting all the potential parameters and measuring their impact on the utility of the clients.

While some progress has been made in terms of contract theory, state-of-the-art methods to select contract groups have not investigated all the potential parameters of the clients. Kang et al. (2019b) demonstrate that the number of contract groups impacts the performance, such that it would be important to investigate different clustering methods. Additionally, this technique does not address the issue of outliers and of clients in low-value groups which might be discriminated unfairly.

Once the clients have a value assigned to them, the server needs to interpret the values to perform the client selection. Thus, the next dimension on our taxonomy is Value Interpretation.

## 3.6 Value interpretation

At the end of the client valuation step, each client has been assigned a value on the server's side. Based on this value, the server selects the clients that will be part of the next training round.

The clients' values are typically interpreted as rankings, probabilities or weights. When the values are interpreted as a **ranking**, the server selects the top $n$ clients with the highest values (Nishio & Yonetani, 2019; Wang et al., 2020; Cho et al., 2022b). If they are interpreted as a **probability** to be selected, the clients are chosen according to such a probability (Goetz et al., 2019; Mohri et al., 2019; Nagalapatti & Narayanam, 2021). Finally, the values might be considered to be **weights** that are applied to the clients' results yielding a weighted sum of client data in the aggregation (Song et al., 2019; Li et al., 2020).

In addition to these three approaches to Value Interpretation, scholars have proposed alternative methods to interpret the clients' data. Chen et al. (2018) propose using a **threshold** to be applied to the client values, which results in a dynamic number of selected clients in each training round. In their experiments, they report that fewer clients (and hence less communication) are needed as the training progresses, and the training reaches the same accuracy $5\times$ faster and with $10\times$ less communication than the baseline. Kang et al. (2019a) keep a reputation score of the clients and only select the clients with a reputation above a pre-defined threshold. They report that increasing this threshold and hence selecting fewer clients boosts the accuracy of the trained model.

Other scholars have proposed **clustering** the clients according to different criteria (e.g. communication costs, available resources, data characteristics) and only a few representatives from each cluster are selected and shared with the server. All the clients in a cluster are treated in the same way from the server's perspective. Examples include Dennis et al. (2021) who select a reduced number of clients from each cluster of similar clients and report 35% less variance in the clients' final test accuracy when compared to random selection, indicating a possible direction towards Good-Intent Fairness; and Zhou et al. (2021) who cluster the clients according to the communication cost with the server.

Note that several methods proposed in the literature leave space for *exploration* in the process of Value Interpretation. We denote these methods with the symbol $\mathcal{E}$ in Table 1. In this case, the server selects a subset of the clients according to their value and leaves a predefined number of slots open to add clients which are selected randomly. For example, Goetz et al. (2019) select $|\mathbb{S}^t| - \epsilon$ clients based on probability and then fill the rest $\epsilon$ clients by random sampling. With a loss-based value generation function, their $AFL_G$ method achieved a 2% area under the curve (AUC) performance increase while needing $30 - 70\%$ fewer epochs to reach this performance compared to a random selection of clients.

### Future work

The final number of participating clients has a impact on the training process. In most cases, this number is defined based on the available resources, such as energy, communication bandwidth or incentive payouts. The Value Interpretation step provides a method to select the number of participating clients depending on such constraints. However, many Value Interpretation approaches require the definition of parameters, such as thresholds or the number of clusters. Thus, automatically identifying the optimal number of participating clients based on their values is still an open research problem.

Another direction of future work consists of investigating further the exploration-exploitation trade-off in client selection. While randomness (i.e. maximal exploration) gives a chance to include unseen clients and improves selection fairness, it also reduces the effectiveness of the selection method. Finding the right balance in the exploration-exploitation spectrum is an therefore a valuable research question to pursue.

In addition to the six previously described dimensions for client selection in FL, we would like to highlight the importance of establishing benchmark datasets and experimental frameworks to facilitate the reproducibility

Table 2: Commonly used datasets in client selection experiments in FL. Clients are either split by classes, or more naturally along a feature of the data–for example, by writers of social media posts.

| Dataset | Split | Type | Methods |
|---|---|---|---|
| MNIST (LeCun, 1998) | Classes | Image | S-FedAvg, FAVOR, CI-MR, FMore, CBIM, |
| FashionMNIST (Xiao et al., 2017) | Classes | Image | pow-d, FAVOR, FedCS, q-FFL, $AFL_M$, FMore, DDaBA |
| EMNIST (Cohen et al., 2017) | Classes | Image | FD |
| FEMNIST (Caldas et al., 2018a) | Writers | Image | $k$-FED, DDaBA |
| CIFAR10 (Krizhevsky, 2009) | Classes | Image | FL-CIR, FAVOR, FedCS, FD, FMore, DDaBA |
| Shakespeare (Caldas et al., 2018a) | Roles | Text | $k$-FED, q-FFL |
| Sent140 (Go et al., 2009) | Writers | Text | q-FFL |
| Reddit (multiple variations) | Writers | Text | q-FFL, Oort, $AFL_G$ |
| UCI Adult Dataset (Blake, 1998) | PhD or not | Tabular | q-FFL, $AFL_M$ |
| London Low Carbon (Marantes & Openshaw, 2012; Schofield et al., 2015) | Households | Timeseries | (Savi & Olivadese, 2021; Briggs et al., 2021) |

of the proposed models and enable the development of comparative analyses. In the next section, we provide a summary of the most commonly used datasets in the literature of client selection in FL.

## 4   Datasets and benchmark experiments

Given the distributed nature of FL and its focus on privacy protection, it is difficult to produce and share realistic open-access Federated Learning datasets.

Thus, most of the experiments reported in the FL literature have been performed on *artificial* FL datasets, generated from well-known benchmark machine learning datasets. Table 2 includes a summary of these benchmark datasets and their characteristics.

Note that creating an *artificially distributed* dataset from an originally centralized one requires designer choices to be made. In the current client selection literature even if two papers use the same dataset, the results are in most cases not comparable due to different choices, such as a different distribution of the data. Moreover, the models that are used (e.g. deep neural networks) vary in different evaluations, adding another layer of difficulty to make comparisons. Given the low reproducibility of current benchmarks, a fair comparison between methods requires the researchers to re-implement and tune each of the relevant past works. The taxonomy proposed in this paper helps to identify the most important methods and their characteristics to enable such a comparison.

In general terms, we find two major approaches to create *distributed* datasets for client selection in FL. The first approach generates clients *based on the target classes* of a classification dataset. With this method, researchers are able to manipulate the non-IID nature of the clients yet the dataset will inherit –potentially unknown– dependencies. For example, in the case of hand-written digits datasets, it is known that there are multiple samples from the same writer. An ideal dataset to simulate a realistic client selection experiment would need to have a large amount of non-IID clients. Unfortunately, this approach does not seem to satisfy such a condition.

In the second approach, the clients are generated *based on a feature* of the samples. In this case, the feature may be the writer of the digits in the MNIST datasets or the author of posts in social media platform datasets –such as, Sent140, Reddit listed on Table 2, or StackOverflow (Reddi et al., 2020). Cross-silo FL scenarios might be simulated by using multiple publicly available datasets and assigning one dataset to each of the clients.

In summary, generating FL datasets from known benchmark datasets does not accurately represent the FL problem at hand, yet obtaining real FL datasets is challenging. We believe that there might be an alternative path moving forward. There are fields with historically available distributed datasets where privacy concerns are emerging. In these cases, the existing data could be leveraged to propose privacy-preserving FL approaches. One of such fields is residential energy consumption. Given current global energy market trends, building accurate predictions of energy consumption will be increasingly relevant both for consumers, producers and energy distributors. With the adoption of smart meters, it is possible to collect detailed data on the consumers' side. In fact, there are several energy datasets available, such as the London Low Carbon project data (Marantes & Openshaw, 2012; Schofield et al., 2015). However, analyzing such sensitive data centrally has clear privacy consequences (Vigurs et al., 2021). Thus, there is an increased need and interest towards FL techniques applied to this use case (Savi & Olivadese, 2021; Briggs et al., 2021). Other domains that could also benefit from FL include self-driving cars, smart city applications, and wearable IoT devices.

Moreover, in many domains there might be several sources of data available to tackle a specific problem. Following the example of the energy consumption prediction problem, in addition to the energy consumption patterns, there are household (Schofield et al., 2015; Wilson, 2014) and weather datasets. Research on model-agnostic FL would enable building models that leverage datasets of different nature across clients.

Beyond using non-FL, pre-existing datasets, there are ongoing efforts to develop specific benchmark datasets for FL. First, we would like to highlight that OpenMinded[1] has started an initiative to build a network where researchers can access distributed data for FL while preserving privacy. Second, the LEAF framework by Caldas et al. (2018a) collects 6 datasets and defines a specific split of the data designed for federated learning.

Regarding baseline models, the reviewed client selection research typically includes the FedAvg algorithm by McMahan et al. (2017) as the baseline. While this offers a much needed standardization in the experimental evaluations, there are more recent federated optimization algorithms that we believe should be considered as baselines, such as FedAdam and FedAdagrad (Reddi et al., 2020). We would like to emphasize the importance for scientists to embrace an open science approach, sharing both the data and the code of newly proposed models. We would also encourage the community to leverage rapidly maturing Federated Learning frameworks –such as, TFF[2], PyShift[3], and Flower[4]– to ease the reproducibility of the results and enable the integration of novel client selection strategies into other parts of the federated pipeline, such as FL optimizers, or differential privacy frameworks.

## 5 Conclusion

As an emergent field, documenting, properly evaluating and comparing novel Federated learning methods is a complex task. In this paper, we have provided an overview of the most notable works in client selection in FL. We have proposed a taxonomy to help researchers identify and compare previous approaches and to support practitioners and engineers in finding the most suitable method for their task at hand. Our taxonomy is composed of 6 dimensions to characterize client selection methods: Policies, Termination Condition, Client Characteristics, Shared Information, Value Generation and Value Interpretation.

The Policies, Client Characteristics, Shared Information and Value Generation help identify the most suited method to satisfy the requirements and goals in a specific scenario. Value Interpretation and the Termination Condition are important to characterize the evaluation and enable the comparison of existing methods.

We have also outlined potential lines of future research regarding each of the six dimensions of the proposed taxonomy and highlighted the need for FL benchmark datasets and models to accelerate progress and ease the comparison of proposed approaches in this field. We hope that the proposed taxonomy will prove to be helpful in this regard.

---

[1]OpenMinded: The Medical Federated Learning Program. Accessed: 2022-06-14, `https://openmined.hubspotpagebuilder.com/medical-federated-learning-program`

[2]`https://www.tensorflow.org/federated`

[3]`https://github.com/OpenMined/PySyft`

[4]`https://flower.dev/`

**Acknowledgements**

G.D.N. and N.O. have been partially supported by funding received at the ELLIS Unit Alicante Foundation from the Regional Government of Valencia in Spain (Generalitat Valenciana, Conselleria d'Innovació, Universitats, Ciència i Societat Digital, Dirección General para el Avance de la Sociedad Digital). G.D.N. is also funded by a grant by the Banco Sabadell Foundation. N.Q. has been supported in part by a European Research Council (ERC) Starting Grant for the project "Bayesian Models and Algorithms for Fairness and Transparency", funded under the European Union's Horizon 2020 Framework Programme (grant agreement no. 851538).

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

# A  Highlighted client selection algorithms

In this section, we provide a summary of the client selection algorithms included in Table 1 and discussed in this paper. These methods have been selected because of their relevance in the field and/or because they represent a specific category of our taxonomy.

**Baseline selection strategies**

The two most commonly used baseline client selection strategies are EqRep and EqW, presented below.

**EqRep$_{\text{baseline}}$**

Equal representation of clients (EqRep$_{\text{baseline}}$): with a total of $N$ clients and $|\mathbb{S}^t|$ selected clients in round $t$ of the training, each client has a $p = \frac{|S^t|}{N}$ probability to participate in the round. This approach is often referred to as *random* client selection. Its motivation is to have all clients participating equally. The original FedAvg method use EqRep$_{\text{baseline}}$ to reduce the number of participating clients.

**EqW$_{\text{baseline}}$**

Equal weights of data samples (EqW$_{\text{baseline}}$): the probability $p_i$ of participation for client $c_i$ is proportional to the fraction of data that the client has access to. Thus, to compute $p_i$ the server needs to know the clients' dataset size, $|\mathbb{D}_i|$, $\forall i = 1, ..., N$. The motivation is to achieve equal representation for all data samples in the training.

**Methods motivated by improving training efficiency**

**AFL$_{\text{G}}$**

In Active Federated Learning (AFL$_{\text{G}}$)[5] the client $c_i$ has a valuation in training round $t$, $v_i^t$, based on its reported loss (Goetz et al., 2019). If a client participates in training round $t$, its valuation is updated, otherwise it remains the same as in the previous training round $t-1$, $v_i^t = v_i^{t-1}, c_i \notin \mathbb{S}^t$. All the clients start with a negative infinite valuation, $v^t = -\infty, \forall c_i \in \mathbb{K}$. The value interpretation is a probability function based on this value with additional clients selected for exploration. The motivation is to achieve the same performance as the baseline client selection algorithms but with fewer training epochs.

**pow-d**

In Power-of-Choice Strategy (pow-d), first a candidate set ($\mathbb{A} \subset \mathbb{K}$) is selected based on the fraction of their data, $|\mathbb{D}_i|$ (as in EqW$_{\text{baseline}}$). Then the clients in $\mathbb{A}$ compute their local losses and report them back to the server. The clients with the highest losses are selected $\mathbb{S}^t \subset \mathbb{A}$ (Cho et al., 2022b). This method improves both the convergence rate and the accuracy when compared to random selection (EqRep$_{\text{baseline}}$).

**S-FedAvg**

In the Shapley-value based federated averaging (S-FedAvg) method proposed by Nagalapatti & Narayanam (2021), the server uses a verification dataset ($D_V$) to determine the clients' contributions using Shapley values. Then, it selects the clients with a probability proportional to their contribution value. This method reaches higher accuracy than FedAvg.

**$k$-FED**

In $k$-FED, a local k-mean clustering is proposed to generate a compressed data description (Dennis et al., 2021). Then, the cluster centroids in each client, $\mathbb{G}_i, c_i \in \mathbb{K}$, are sent to the server. The server does not include in the training clients with similar cluster centroids. This approach is evaluated with a combination of pow-d and the results show that it can boost the training convergence even further.

**LAG**

---

[5]AFL refers to Agnostic Federated Learning (Mohri et al., 2019), Active Federated Learning (Goetz et al., 2019) and Anarchic Federated Learning (Yang et al., 2022). We separate them using the initials of the first authors, AFL$_{\text{M}}$, AFL$_{\text{G}}$ and AFL$_Y$ respectively.

The Lazily Aggregated Gradient (LAG) (Chen et al., 2018) method stores previous updates of the clients and predicts their future performance using a Lipschitz-smoothness ($\mathcal{L}$) estimator. In this approach, the updates are computed only on valuable clients as decided by the client or the server. Early stopping is possible if none of the clients sends updates. The motivation for the development of this method is to reduce the communication between clients and the server.

### FL-CIR

Federated learning with class imbalance reduction (FL-CIR) (Yang et al., 2021) proposes a multi-armed bandit method to select a client set with minimum class imbalance. Each client represents an arm, and the selected clients are a super-arm (set of arms). The reward of the super-arm is computed based on the results of the current training round. The motivation is to reduce the impact of non-IIDness in the data and hence improve the overall model's accuracy.

### FAVOR

Optimizing Federated Learning on Non-IID Data with Reinforcement Learning (FAVOR) Wang et al. (2020) formulates the client selection problem as a RL task, where the server acts as an agent, the state $s^t = \boldsymbol{\theta}^t, \boldsymbol{\theta}_1^t, ..., \boldsymbol{\theta}_N^t$ summarizes the current parameters in each client, the action consists of selecting a client for the next training round and the reward is the sever-side accuracy. The server has a local validation set ($D_V$) to evaluate the clients' models. The method's effectiveness is demonstrated with experiments where the target accuracy is reached with fewer communication rounds than previous methods.

### Other global constrains

In addition to improving the training efficiency, several research works have explored other global constrains to guide the client selection process.

### FedCS

Nishio & Yonetani (2019) proposed one of the first client selection methods called Federated Learning with Client Selection (FedCS). The clients report their resource requests to the server in the form of an estimated compute time $\mathcal{T}$. The global constrain is defined as a maximum compute time per client. Thus, the server selects the clients that fulfill such a constrain.

### AFL$_M$

Agnostic Federated Learning (AFL$_M$), Mohri et al. (2019) aims to minimize the maximum loss of the clients' performance (good-intent fairness). The server updates a $\lambda$ distribution function in every training round based on these local losses, and $\lambda$ is used to reweigh the samples or to select clients that match a target data distribution.

### q-FFL

q-Fair Federated Learning (q-FFL) aims to achieve a uniform accuracy across all clients Li et al. (2020). First, the server select clients according to a uniform probability. Then, the server reweighs the client parameters based on the Lipschitz constant ($\mathcal{L}$) of the local loss functions.

### Oort

In Oort, Lai et al. (2021) utility scores are used to rank the clients and the server selects the top-K performers. The utility function depends on the local loss and the training time and it is given by: $Util(i) = |\mathbb{B}_i|\sqrt{\frac{1}{\mathbb{B}_i}\sum_{k\in\mathbb{B}_i}L(k)^2} \times \frac{\mathcal{T}}{\mathcal{T}_i}^{\mathbb{1}(\mathcal{T}<\mathcal{T}_i)\times\alpha}$, where $\mathbb{B}_i \subset \mathbb{D}$ is a subset of samples in client $i$; $\mathcal{T}$ and $\mathcal{T}_i$ are the global and local needed training time, respectively; $\mathbb{1}(x)$ is 1 if $x$ true, 0 otherwise, and $\alpha$ is a hyperparameter. Oort also includes an exploration component to increase the data diversity.

### FLAME

FLAME: Federated Learning Across Multi-device Environments (FLAME$_C$)[6] builds energy profiles for each client and clients have a maximum allowed budget of energy consumption. The clients compute their utility scores using the training loss ($L$), the energy consumption ($E$) and the required computational time($\mathcal{T}$). The server selects the clients with the largest overall utility. In their experiments, even though clients have limited energy and therefore, limited participation (Cho et al., 2022a), they achieve.... add results

### DDaBA

Dynamic Defense Against Byzantine Attacks (DDaBA) (Rodríguez-Barroso et al., 2022) tries to identifiy adversarial clients based on the local update's accuracy on the server-side validation set $\mathbb{D}_V$. It flags potentially harmful clients when the accuracy of their local update changes dramatically between rounds. Experiments show that this approach is effective against byzantine attacks.

### Client inclusion policies

Regarding client inclusion policies, we highlight two methods proposed in the literature.

### LT-FL

Loss Tolerant Federated Learning (LT-FL) (Zhou et al., 2021) classifies the clients into two categories, depending on whether they have enough resources to retrieve lost communication packages or not. The clients in the first group are expected to send an updated model to the server in all cases. However, the weights of the clients from the second group may be set to zero if their packages get lost.

### FD

Federated Dropout (FD) (Caldas et al., 2018b) use model compression (via dropped neurons) to reduce the size of the clients. The dropped neurons are not randomized but computed in such a way to have the same smaller matrix dimension in each client. Moreover, the server is able to map them back together to their original model size. This technique increases the server-client communication efficiency while keeping the original prediction accuracy. Later work (Diao et al., 2020; Liu et al., 2022) use the same idea to allow for heterogeneous model sizes in clients with different capabilities.

### Incentive mechanisms

Finally, we include a summary of the three highlighted methods that propose different incentive mechanisms.

### CI-MR

In Contribution Index Multi Round Reconstruction (CI-MR), all clients perform their local training and send the results back to the server which weights them depending on the size of the local datasets in each client, similarly to EqW$_{baseline}$. At the end of the training, the server uses Shapley values to determine the data quality of the clients to define the payment of their incentive (Song et al., 2019).

### FMore

FMore: an Incentive Scheme of Multi-dimensional Auction (FMore) (Zeng et al., 2020) is an incentive mechanism where the server shares a scoring function (Value Generation function) with the clients, the clients compute their utility scores based on their local resources and offer a bid with a payment proposal. Based on this information, the server selects the clients for that round of training.

### CBIM

Contract-Based Incentive Mechanism (CBIM) (Kang et al., 2019a) uses contract theory to select the right cluster of clients based on the clients' utility scores. It also keeps track of the reputation of the clients via a blockchain mechanism.

---

[6]There are at least 3 FL papers with the FLAME acronym. We include the initial of the first author as a subscript to differentiate between them.

