# OpenReview forum: "A Snapshot of the Frontiers of Client Selection in Federated Learning"
_TMLR — Accepted by TMLR_

### Review · Reviewer_AwBe · 2022-09-28

**Summary Of Contributions:**

This paper provides a systematic study of client selection algorithms in federated learning. It proposes a taxonomy to organize and classify existing selection algorithms and compare their pros and cons.

**Requested Changes:**

N/A

**Strengths And Weaknesses:**

Strengths:
1. The survey is complete. It covers most selection algorithms to date.
2. The proposed taxonomy is detailed and catches important features in FL.

---

> ### Author Response · Authors · 2022-11-09
> **Answer to Review of Paper414 by Reviewer AwBe**
>
> Thank you very much for your positive feedback regarding our manuscript!

---

### Review · Reviewer_Tz1r · 2022-10-17

**Summary Of Contributions:**

The paper provides an overview of common works on client selection in Federated Learning. The review is thorough and provides a taxonomy to classify the papers and makes observations on lack of comparisons and common benchmarks.

**Broader Impact Concerns:**

The work has a strong potential to have an impact on application of FL. Engineers and researchers might use this work to decide on client selection methods. The quality of writing and presentation are also high.

**Requested Changes:**

Addressing above questions would significantly strengthen the submission.

**Strengths And Weaknesses:**

I liked a lot a thoughtful analysis of each part in the taxonomy and ideas for the future work which I mostly agree with.

The split between client and server level is also useful and very practical, e.g. the engineer/researcher might decide which part to implement.

However, I'd ask for some clarification on the following parts:

1. **Privacy**:
    - What are the privacy definitions that are used in the paper? Client selection is inherently privacy-sensitive operation which might leak different attributes of user data including metadata such as user participation patterns or reaction to incentives (as one of the client selection methods).

    - I'd maybe dedicate a part in the background on a definition for privacy loss or specify at each discussion what exactly the conversation is about. Even the basic notion, i.e. who is the adversary and that learning more about the user data than before participating in FL is privacy leakage (i.e. following differential privacy definition).

    - After the definition it will be really interesting to point that some of the server-side methods specifically target the privacy leakage to rank the users.

2. **Limitations with other FL techniques**
    - Differential Privacy assumes randomized sampling (with some relaxation) and might limit some of the methods.
    - Secure Aggregation and Local Differential Privacy are both very popular in FL literature but to me seem like incompatible as the server will have no visibility into user updates (in case of SA) or no ability to analyze the update as it will be similar to noise (LDP).

3. **Trustworthiness**
    - In general, any client-selection will motivate the attacker to game the selection protocol.
    - Note on accuracy: the attacker might provide high accuracy for the update, but still be malicious, e.g. inject a backdoor. Also some clients might provide low accuracy and still be useful (especially in non-iid case).
    - Some defenses might be abused by the attacker (if they manage to abuse client selection protocol). For example, the cited method (Krum by Blanchard et al, 2017) selects clients with the smallest weight update is vulnerable to adaptive attacks. It can be abused by injecting a backdoor with a very small learning rate, and therefore small weight update. The server will then be selecting this client over other benign clients.

4. **Datasets**
    - I'd emphasize even further that performing client selection on MNIST-like datasets is not the most convincing experiment.
    - Maybe to balance it out, it's also useful to acknowledge that achieving convergence on Reddit is a computationally expensive task.
    - Another popular dataset in FL not referenced yet is StackOverflow, might be useful for the researchers.

---

> ### Author Response · Authors · 2022-11-09
> **Answer to Review of Paper414 by Reviewer Tz1r**
>
>
> We would like to thank the reviewer for appreciating the contributions of our paper. We are grateful for the constructive and valuable feedback which we have included in the revised version of the manuscript (marked in red font). Special thanks for the very appropriate suggestion to include two key challenges in client selection in FL: the importance of the privacy implications and the vulnerability against attacks to the client selection strategy. These challenges have been omitted by previous work and we certainly believe that it would be important to address them in future work. The revised version of the manuscript includes new paragraphs related to these challenges.
>
> **Privacy Definition and Privacy Loss Definition**
>
> We have added paragraphs related to privacy both in the Introduction and in the Shared information section (3.4) of the paper. Thank you for the suggestion!
>
> **Limitations of other FL techniques**
>
> Thank you for pointing out this limitation. It supports the need for the Shared Message component of our taxonomy. An approach to leverage secure aggregation and differential privacy would be to apply them on the model weights, but send the additional metadata visible to the server. However, this strategy would not be compatible with methods that rely on server-side validation. However, approaches, such as FedCS could still be used. We have added a discussion regarding this issue in Section 3.4 of the manuscript.
>
> **Trustworthiness**
>
> Thank you very much for the very valuable comment, which we have added to Section 3.1.1 Resilience of the paper.
>
> **Datasets**
>
> We included your suggestions in Section 4, page 16 of the manuscript. Thanks so much!

---

> > ### Comment · Reviewer_Tz1r · 2022-11-22
> > **reviewer response**
> >
> > Thank you for addressing the comments, I am happy with the changes.

---

### Review · Reviewer_wdU1 · 2022-10-23

**Summary Of Contributions:**


This paper gives a snapshot of some recent advances on the client selection problem in federated learning. It provides a taxonomy of different methods, summarizes existing works, and discusses future research questions in client selection. This submission may help both researchers and practitioners get a better view of the research landscape, apply current techniques to the tasks at hand, and inspire future research and adoption.


**Broader Impact Concerns:**

None.

**Requested Changes:**

* Figure 1: Some algorithms only need one round of communication to perform client selection and get the model updates from selected clients. E.g., they dynamically maintain some (maybe stale) information (e.g., client gradients, client losses) on the server side (e.g., Towards Understanding Biased Client Selection in Federated Learning, Diverse client selection for federated learning via submodular maximization). Hence, this figure cannot represent a class of client selection methods.

* " Furthermore, it limits the ability of practitioners to apply the most suitable of the existing FL techniques to a specific real-world problem, because it is not clear what the state-of-the-art is in the scenario required by their application": this question proposed by this submission is not addressed or investigated with evidence.

* Section 3.1.1: The motivation part needs to be better organized/structured. For example, what are the differences between 'training efficiency' and 'limited complexity'?

* Sections 3.1.2 'Client inclusion policies' are more of approaches rather than motivation for client selection.

* Section 3.1.3 has much overlap with other subsections. For example, improving 'global constraints' (training efficiency, fairness) naturally has incentives for clients. Moreover, Section 3.1.3 describes many methods on the problem of client incentive itself, but it is not clear how client selection can serve as incentivization mechanisms, and what previous works have done on this front. It is clear that client selection is related to incentive mechanisms, but this subsection is only talking about incentive mechanisms, which is disconnected from the 'motivation' of client selection.

* Future work after Section 3.1.3: Several proposed future motivations are not convincing. The paragraph on privacy is arguing that future research should consider privacy as a metric when designing client selection methods, but for this point to be future work for 'motivation', the paper needs to discuss why different client selection methods can improve privacy/utility tradeoffs (which I am not sure to what extent this is true).

* In terms of reliability (bottom of Page 7): (a) It is not clear what reliability means in this paragraph. From the last sentence regarding clustering, it seems that reliability can mean representation fairness, which has already been discussed in existing works.
(b) Moreover, selecting clients from each cluster has been explored in previous works (e.g., Heterogeneity for the Win: One-Shot Federated Clustering), which shouldn't be included as future work.

* The last two paragraphs of Section 3.1: The current text talks about future directions in improving client selection strategy and evaluation of incentive mechanisms, but not necessarily the future work of the 'motivation for client selection'.

* Section 3.2: (a) The problem of termination conditions broadly exists in iterative methods (not necessarily machine learning). I do not think the two conditions discussed (fixed rounds and certain accuracy) are conditions that are particularly interesting for the problem of client selection. Hence, it is not necessary to include them in the current manuscript. (b) The proposed termination conditions (based on privacy budget and client-centric) are reparameterization of the existing conditions. For example, to satisfy a privacy budget, one practical implementation is to calculate the number of rounds under a privacy loss, and run the fixed number of rounds.


* Section 3.3: In future work, the arguments made in the first paragraph (considering client states) have been proposed before (A Field Guide to Federated Optimization).In addition, using over-selection to address client dropout is actually a standard and default practice of federated methods, not future work.

* Section 3.5: I do not think it is necessary to list the exact numbers in previous works, as each paper has different experiment setups and it is impossible to reason about those numbers. This applies to other parts of the submission as well.

* Section 4: The discussions are very generic to FL, instead of client selection.




**Strengths And Weaknesses:**

Strengths:
* This submission focuses on an important problem of client selection in federated learning. As there are more and more works in this area, it provides some centralized discussions of the current research efforts.
* It abstracts the client selection problem into a simple framework which covers most of the existing methods (though with caveats as described later), and provides a taxonomy of different methods, considering different settings (cross-silo/cross-device) and practical constraints (systems and algorithmic requirements).
* It discusses future works organized into different categories along with the taxonomy of existing work.

Weaknesses:
* While this is a survey paper focusing on client selection, lots of works discussed are not proposing new client selection methods, but just related to client selection (sometimes very loosely). Large portions of text are disconnected from client selection, but just enumerating existing works on some topics of federated learning (see requested changes below). For example, a large subset of methods listed in In Table 1 and Table 2 are not any specific client selection methods. The technical details of those methods are heavily discussed including their experimental results and exact numbers), but it would be better to focus more on their connections with client selection.
* One important question—which client selection methods are generally good empirically—remains unanswered. It is still not clear how much better complicated client selection strategies really are compared with random selection.
* Many issues and future works discussed are generally interesting problems in federated optimization/learning itself, but not closely tied to the problem of client selection.
* Technical content, organization, and writing need to be largely improved.

My other concerns are listed in the requested changes.

---

> ### Author Response · Authors · 2022-11-09
> **Answer to Review of Paper414 by Reviewer wdU1 #1**
>
> We are grateful to the reviewer for the constructive and detailed feedback which has been instrumental to improve our manuscript. Following your suggestions, made the necessary changes in the revised version of the paper (marked with red font). Below, we provide a point-by-point response to your questions and suggestions.
>
> **Client selection vs Generic FL problems**
>
> In our paper, we have considered FL approaches that include a client participation management strategy. We believe that this inclusive perspective is important to help understand the big picture related to client management in FL. All the methods that are listed in Table 1 (and 2) satisfy that condition.
>
> It would be great if you could please let us know which specific methods in Tables 1 and 2 do not include a client participation/selection strategy so we can update the tables accordingly. We are unsure of what methods you might refer to. Note that weight value interpretation methods -where the clients generally participate in every training round-- may reduce the weight of clients to zero, which would be equivalent to not including them.
>
> **Which client selection methods are generally good empirically remains unanswered**
>
> This is a good question. In fact, the lack of a systematic way to compare methods was the motivation for our paper. We believe that the proposed taxonomy, together with some of the proposed lines of future work, will be helpful to address your question.
>
> In any case, note that several of the methods that are discussed in the paper --such as pow-d, k-FED and AFL-- report clear efficiency gains when compared to random selection. To highlight the performance improvements of client selection methods, we included such improvements when we discuss the methods in the paper.
>
> **Figure 1**
>
> Thank you very much for your valuable suggestion. We have added the $\gamma$ function to the Figure and corresponding text to showcase stateful servers. Thanks to your recommendation, we believe that Figure 1 depicts a more general client selection diagram.
>
> **Motivation related to a lack of clarity on the state-of-the-art for a particular application**
>
> The evidence for this question is shown by the lack of FL benchmark datasets and of consistency in how experiments and results are reported in the field, which severely limits the reproducibility of the results. Your second to last bullet point seems to reach a similar conclusion “Section 3.5: I do not think it is necessary to list the exact numbers in previous works, as each paper has different experiment setups and it is impossible to reason about those numbers. “ This is precisely one of the key drivers for the proposed taxonomy.
>
> **Section 3**
>
> Based on your extensive feedback regarding Section 3 of the paper, we have renamed the “Motivation” component of our taxonomy as “Policy”, given that the term “motivation” was confusing. Thank you very much for your very valuable suggestion to change this section of the paper.
>
> **Section 3.1.1**
>
> Thank you for pointing this out. We have merged Training efficiency and Limited complexity in the same subsection.
>
> **Sections 3.1.2**
>
> We believe that changing the element of the taxonomy from Motivation to Policy strengthens the client inclusion’s component of the taxonomy.
>
> **Section 3.1.3**
>
> Policies are not exclusive: a client selection method may implement several policies at the same time, such as global constraints and incentives.
>
> **Future work after Section 3.1.3**
>
> Regarding your questions about privacy, note that we have extended the discussion about privacy following Reviewer Tz1r's suggestion. The revised version of the manuscript clarifies this aspect. Thanks for pointing this out.
>
> **In terms of reliability...**
>
> Thank you for the suggestion. For clarity, we have removed the paragraph from the future work of this section.
>
> **The last two paragraphs of Section 3.1**
>
> With the new terminology, we believe that we address your comment. Thanks!

---

> > ### Author Response · Authors · 2022-11-09
> > **Answer to Review of Paper414 by Reviewer wdU1 #2**
> >
> > **Section 3.2: (a)**
> >
> > We agree with you that termination conditions broadly exist in iterative methods beyond FL. However, we have included the Termination Condition as part of the taxonomy for completion given that it is used by several FL methods, including to limit the participation of certain clients, which is related to client selection.
> >
> > **Section 3.2: (b)**
> >
> > Note that in client selection methods that have a privacy budget, the clients may stop at different times, which is different from a global constraint. If the privacy budget is defined by training rounds, indeed, it means that every client would have a fixed number of training rounds. However, it does not need to be the case.
> >
> > **Section 3.3**
> >
> > The paper cited by the reviewer discuss stateful clients in general for FL. We think it is important to highlight it for Client Selection too. We added reference to the mentioned paper.
> >
> > **Section 3.5**
> >
> > We have provided the exact numbers to highlight the performance gains reported in the client selection literature when compared to baseline methods, such as random selection of clients.
> >
> > **Section 4**
> >
> > We have devoted a Section in the paper to Datasets because the lack of benchmark datasets for client selection in Federated Learning is a big limitation today which prevents a fair comparison between methods, as you rightly mention in your comment regarding Section 3.5. While the listed datasets are not exclusively used in the client selection literature, the client selection methods that are discussed in the paper include them in their experimental results. Thus, they are relevant to the client selection literature. By highlighting this limitation regarding datasets, we hope to steer the community in the direction of producing realistic benchmark datasets for client selection in FL.

---

### Decision · Action_Editors · 2022-12-11

**Recommendation:** Accept as is

**Comment:**

The paper presents a well-defined taxonomy for the important sub-problem of client selection in federated learning. This work will make future research in this area more valuable, interpretable, and comparable. The reviewers were all positive about the paper, and the authors did a good job of addressing the reviewers' comments in the revised manuscript. Overall, this paper is ready to be accepted and I also recommend it for a survey certification.

**Audience:**

TMLR's audience who work on federated learning will be interested in knowing the findings of this paper.

**Claims And Evidence:**

This paper presents a comprehensive survey of client selection in the context of federated learning. The authors provide a useful taxonomy for categorizing different client selection approaches, and they provide a detailed overview of the existing work in this area. The technical content of the paper appears to be accurate, and the authors do a good job of highlighting the contributions and limitations of the existing literature. Overall, this paper is a valuable resource for researchers interested in client selection in federated learning.